# Effect of Antibiotic Susceptibility and *CYP3A4/5* and *CYP2C19* Genotype on the Outcome of Vonoprazan-Containing *Helicobacter pylori* Eradication Therapy

**DOI:** 10.3390/antibiotics9100645

**Published:** 2020-09-26

**Authors:** Mitsushige Sugimoto, Daiki Hira, Masaki Murata, Takashi Kawai, Tomohiro Terada

**Affiliations:** 1Department of Gastroenterological Endoscopy, Tokyo Medical University Hospital, Shinjuku, Tokyo 160-0023, Japan; t-kawai@tokyo-med.ac.jp; 2Division of Digestive Endoscopy, Shiga University of Medical Science Hospital, Otsu, Shiga 520-2192, Japan; 3Department of Pharmacy, Shiga University of Medical Science Hospital, Otsu, Shiga 520-2192, Japan; hirad@fc.ritsumei.ac.jp (D.H.); teradat@belle.shiga-med.ac.jp (T.T.); 4College of Pharmaceutical Sciences, Ritsumeikan University, Kusatsu, Shiga 525-8577, Japan; 5Department of Gastroenterology, National Hospital Organization Kyoto Medical Center, Kyoto 612-8555, Japan; mura05310531@gmail.com

**Keywords:** Vonoprazan, *CYP3A5*, *CYP2C19*, *Helicobacter pylori*, drug resistance

## Abstract

Background: *Helicobacter pylori* eradication containing the potassium-competitive acid blocker, vonoprazan, achieves a higher eradication rate than therapy with proton pump inhibitors (PPIs). Because vonoprazan is mainly metabolized by CYP3A4/5, CYP genotype may affect the eradication rate. We investigated the influence of antibiotic susceptibility and *CYP3A4/5* and *CYP2C19* genotypes on the eradication rates. Methods: A total of 307 Japanese who were genotyped for *CYP3A4* *1/*22, *CYP3A5* *1/*3 and *CYP2C19* *1/*2/*3/*17, and investigated for susceptibility to antimicrobial agents, received vonoprazan-containing regimens: (1) With amoxicillin and clarithromycin as the first-line treatment; (2) with amoxicillin and metronidazole as the second-line treatment; or (3) with amoxicillin and sitafloxacin as the third-line treatment. Results: The eradication rate was 84.5% (95% confidence interval [CI]: 78.9–89.1%) using first-line, 92.6% (95% CI: 82.1–97.9%) using second-line and 87.5% (95% CI: 73.1–95.8%) using third-line treatment. Infection with clarithromycin-resistant strains was a predictive factor for failed eradication (odds ratio: 5.788, 95% CI: 1.916–17.485, *p* = 0.002) in multivariate analysis. No significant differences were observed in the eradication rate of regimens among *CYP3A4*, *CYP3A5* and *CYP2C19* genotypes. Conclusions: Genotyping for *CYP3A4* *1/*22, *CYP3A5* *1/*3 and *CYP2C19* *1/*2/*3/*17 before vonoprazan-containing eradication treatment may not be useful for predicting clinical outcomes.

## 1. Introduction

In 2013, the Japanese health insurance system began covering eradication treatment for all *Helicobacter pylori*-positive patients [1]. However, eradication rates of first-line therapy, twice-daily dosing (bid) with a proton pump inhibitor (PPI), amoxicillin (750 mg), and clarithromycin (200 mg or 400 mg) for 7 days, have decreased to approximately 65% due to a > 35% prevalence of clarithromycin-resistant *H. pylori* strains in Japan [1,2]. The cure rate of PPI-containing triple therapy is affected by several factors, including antibiotic susceptibility (e.g., clarithromycin, amoxicillin, and metronidazole) [1,2,3], insufficient acid inhibition during eradication treatment (e.g., *CYP2C19* genotype, dose of drug, treatment schedule and type of acid-inhibitory drug) [3,4,5], the environment (e.g., smoking), poor adherence to medication and infection of *H. pylori* strain with low virulence activity (e.g., *cagA*-negative, *vacA* s2 genotype and *dupA*-negative strains) [4,6]. Previously, we showed that an intragastric pH > 4 must be maintained for 24 h and that the 24-h pH should be higher than 6.0 for successful eradication in the first-line triple therapy [5]. However, because it is not possible to maintain pH > 4.0 for 24 h in all patients using triple therapy with a standard dose of PPI bid, [7,8] treatment failure is often observed in such an insufficient patients [5,9,10]. A recent study reported that vonoprazan 20 mg inhibits H^+^/K^+^-ATPase activity at a 400-fold lower dose than lansoprazole 30 mg at pH 6.6 [11]. Therefore, vonoprazan-containing eradication therapy appears to have a higher eradication rate than PPI-containing regimen [12,13,14,15,16]. 

Because PPIs, such as omeprazole, lansoprazole and rabeprazole, undergo extensive hepatic metabolism by *CYP2C19* [17], differing plasma PPI levels and intragastric pH values after administration of PPIs among *CYP2C19* genotypes can be a clinical problem [7,18,19,20]. In general, the effect of *CYP2C19* genotype on outcome of PPI-containing eradication treatment cannot be ignored, especially in *CYP2C19* extensive metabolizers (EMs). In contrast, vonoprazan is metabolized to its inactive form mainly by CYP3A4/5 and partially by CYP2C19, making the effects of vonoprazan less affected by *CYP2C19* genotype than PPIs [21]. However, although it is expected to associate with the outcome of vonoprazan-containing eradication therapy and *CYP3A4/5* genotype, it is unclear whether *CYP3A4/5* and *CYP2C19* genotype influence the outcome of vonoprazan-containing therapy [22].

It is important to identify any factors that may influence the clinical outcome of vonoprazan-containing eradication therapy and to determine optimal alternative vonoprazan-containing regimens. Here, we assessed antibiotic susceptibility and the influence of *CYP3A4/5* and *CYP2C19* genotype on the clinical outcome of first-line, second-line and third-line vonoprazan-containing triple eradication therapies in Japanese.

## 2. Materials and Methods 

We retrospectively investigated a total of *H. pylori*-positive patients treated with eradication therapy after upper gastroduodenal endoscopy at the Shiga University of Medical Science Hospital from April 2015 to December 2019 (Table 1). We excluded patients who had no informed consent, no result of eradication outcome, no detailed clinical information including CYP2C19 genotype and had more than three time of eradication history (Figure 1). 

The study was conducted in accordance with the guidelines of the Declaration of Helsinki. Approval for the study protocol was obtained in advance from the Institutional Review Board of the Shiga University of Medicine Science. Written informed consent was obtained from each patient who underwent endoscopy. The information about this study was available online of Shiga University of Medical Science Hospital, and the participants gave their informed consent.

### 2.1. Treatment Regimen

Patients were treated with vonoprazan 20 mg bid and a combination of two antimicrobial agents, amoxicillin (750 mg, bid) and clarithromycin (200 mg, bid) as first-line treatment (*n* = 213); amoxicillin (750 mg, bid) and metronidazole (250 mg, bid) as second-line treatment (*n* = 54); or amoxicillin (500 mg, qid) and sitafloxacin (100 mg, bid) as third-line treatment (*n* = 40) for 7 days. 

#### 2.1.1. CYP2C19, CYP3A4 and CYP3A5 Genotyping

Genomic DNA was extracted from the blood (DNA Extract All Reagents, Applied Biosystems, Foster City, CA, USA). Genotyping was performed using a single-nucleotide polymorphism genotyping assay (StepOnePlus^TM^, Applied Biosystems) in a real-time polymerase chain reaction system. We evaluated polymorphisms in *CYP3A4*22* (rs35599367, C>T) and *CYP3A5*3* (rs776746, G>A). To classify each subject, genotyping was performed to identify the *CYP2C19* wild-type gene and three mutant alleles, *CYP2C19 *2* (rs4244285, A/G)*, *3* (rs-4986893, G/A) and **17* (rs12248560, A/C/T). 

#### 2.1.2. Esophagogastroduodenoscopy

The six grades of the Kimura-Takemoto gastric atrophy classification were used: Closed (C)-I, C-II, C-III, and Open (O)-I, O-II, and O-III [23]. Patients were also scored based on the severity of atrophy, intestinal metaplasia, hypertrophy of gastric folds, and diffuse redness according to the Kyoto classification of gastritis [24].

#### 2.1.3. Infection Status and Measurement of Antibiotic Resistance

Infection status was evaluated based on findings from three tests: An anti-*H pylori* IgG serological test, a rapid urease test, and a culture test. Patients were diagnosed as being positive for *H. pylori* infection when positive results were obtained in at least one of the three tests. Eradication success was evaluated using a ^13^C-urea breath test with a cut-off value of 2.5 ‰ at 6–8 weeks after treatment. 

For bacterial culture and antimicrobial sensitivity testing, agar plates were inoculated with biopsy specimens and incubated at 37 °C under microaerophilic conditions. *H. pylori* colonies were subcultured using the agar dilution method to determine the minimum inhibitory concentration (MIC) for amoxicillin, metronidazole, clarithromycin and sitafloxacin. Cut-off MICs used to define resistance were > 1.0 µg/mL for clarithromycin and sitafloxacin and > 8 µg/mL for metronidazole [25]. The cut-off MICs used to define resistance and the absence of sensitivity were > 0.5 µg/mL and > 0.06 µg/mL for amoxicillin, respectively.

### 2.2. Data Analysis

The patients enrolled in this study who were recruited from 2015 to 2017 were overlapped with those in our previous report performed as a preliminary study (*n* =126) [22]. Age is shown as mean ± standard deviation (SD). Eradication rate was evaluated using intention-to-treat (ITT) and per-protocol (PP) analyses and calculated with 95% confidence intervals (CIs). Statistical differences in eradication rates among the regimens and *CYP2C19, CYP3A4*, and *CYP3A5* genotypes were assessed using Fisher’s exact test. All p-values were two-sided, and *p* < 0.05 was considered statistically significant. Calculations were conducted using commercial software (SPSS version 20, IBM Inc.; Armonk, NY, USA).

## 3. Results

Of a total of 353 *H. pylori*-positive patients treated eradication therapy at Shiga University of Medical Science Hospital from April 2015 to December 2019, we excluded 46 patients who had no result of eradication outcome (*n* = 5), no detailed clinical information including CYP2C19 genotype (*n* = 40) and had more than three time of eradication history (*n* = 6) (Figure 1). Of the remaining 307 patients, 213 had no prior eradication history, 54 had undergone one course of eradication treatment (PPI or vonoprazan with clarithromycin and amoxicillin), and 40 had undergone two courses (PPI or vonoprazan with clarithromycin and amoxicillin as first-line treatment and PPI or vonoprazan with metronidazole and amoxicillin as second-line treatment). There were no significant differences in age, sex, body weight, height, or history of smoking and drinking among the first-, second-, and third-line treatment groups (Table 1). 

In patients receiving first-line therapy, the prevalence of the *CYP2C19* genotype was 3.3% (7/213) in ultra rapid metabolizers (UR), 25.8% (55/213) in EM, 52.1% (111/213) in intermediate metabolizers (IM) and 18.8% (40/213) in poor metabolizers (PM) (Table 1). The prevalence of *CYP3A5* polymorphisms was 8.3% for *CYP3A5* *1/*1, 39.3% for *1/*3, and 52.4% for *3/*3 (Table 1). There was no significant difference in the prevalence of *CYP2C19* or *CYP3A5* genotype among patients receiving first-, second- and third-line therapies. In this study, no patients had the *22 mutant allele in *CYP3A4*. 

In the association with eradication time and endoscopic finding, the score for diffuse redness in patients receiving third-line therapy was significantly lower than that in patients receiving first- (*p* = 0.006) and second-line therapies (*p* = 0.022) (Table 1). 

### 3.1. Susceptibility to Antimicrobial Agents

Among patients receiving first-line therapy, 40.6% (56/138) had clarithromycin-resistant strains (Table 1). The proportion of patients showing no sensitivity to amoxicillin and metronidazole resistance was 14.5% (20/138) and 8.0% (11/138), respectively. Among patients receiving second-line therapy, the proportion showing clarithromycin and metronidazole resistance and no sensitivity to amoxicillin was 77.4%, 9.7%, and 32.3%, respectively. 

### 3.2. Eradication Rates in Vonoprazan-Containing Eradication Therapy

The eradication rate was 84.5% (95% CI: 78.9–89.1%) using first-line treatment, 92.6% (95% CI: 82.1–97.9%) using second-line treatment and 87.5% (95% CI: 73.1–95.8%) using third-line treatment (*p* = 0.295) (Table 2). Although the eradication rate in *CYP2C19* UM treated with first-line treatment was lower (57.1%) than that in other genotypes (81.1–90.9%), there was no significant difference among *CYP2C19* genotypes irrespective of whether they received first-, second-, or third-line therapy. Among those who received first-line treatment, the eradication rate in *CYP3A5* *1 allele carriers was 82.1% (95% CI: 69.6–91.1%, 46/56), which is similar to that in the *3/*3 type (84.7%, 95% CI: 73.0–92.8%, 50/59, *p* = 0.344) (Table 2). 

Among those who received first-line treatment, the eradication rate in patients infected with *H. pylori* strains sensitive and resistant to clarithromycin was 91.5% (95% CI: 83.1–96.5%) and 71.4% (95% CI: 57.8–82.7%), respectively (Table 2). The eradication rate in patients infected with strains that were not sensitive to amoxicillin was lower than that in patients infected with amoxicillin-sensitive strains, albeit not significantly so. Likewise, no significant difference in eradication rate was observed between metronidazole-sensitive and -resistant strains. 

In the univariate analysis, infection with clarithromycin-resistant strains (OR: 4.286, 95% CI: 1.628–11.279, *p* = 0.003), endoscopic diffuse redness (OR: 2.868, 95% CI: 1.092–7.530, *p* = 0.032) and total endoscopic score based on the Kyoto classification of gastritis (OR: 1.470, 95% CI: 1.057–2.043, *p* = 0.022) were identified as predictive factors for failed eradication (Table 3). In the multivariate analysis, which examined factors showing a *p*-value < 0.2 in the univariate analysis (sex, endoscopic intestinal metaplasia, endoscopic diffuse redness, clarithromycin resistance, and no sensitivity to amoxicillin), infection with clarithromycin-resistant strains was identified as a predictive factor for failed eradication (OR: 5.788, 95% CI: 1.916–17.485, *p* = 0.002) (Table 3). 

### 3.3. Complications

Among 260 patients, 63 experienced adverse events, including diarrhea (*n* = 19), loose stools (*n* = 18), abdominal pain (*n* = 9) and allergic reaction (*n* = 10) (Table 4). There was no significant difference in the incidence of adverse events among the treatment regimens.

## 4. Discussion

Successful eradication therapy for *H. pylori* infection using acid-sensitive anti-microbial agents, such as clarithromycin and amoxicillin, requires the maintenance of an intragastric pH > 4.0 for 24 h a day [5,9,10]. In this study, we demonstrated that vonoprazan-containing eradication therapy was well tolerated and achieved an eradication rate of approximately 85%, irrespective of prior eradication history (84.5% for first-line, 92.6% for second-line, and 87.5% for third-line therapy). In addition, we examined the influence of genetic variations in drug-metabolizing liver enzymes (i.e., *CYP2C19*, *CYP3A4*, and *CYP3A5*) on the outcomes of vonoprazan-containing regimens and found that there was no significant difference among genotypes. Despite the preliminary nature of this study, owing to its limited sample size, we showed that infection with clarithromycin-resistant strains may be a risk factor for failed eradication using first-line vonoprazan-containing therapy. We therefore recommend the use of alternative first-line therapies in patients infected with clarithromycin-resistant strains.

### 4.1. Eradication Rate of Vonoprazan-Containing First-Line Eradication Therapy

Vonoprazan induces sustained acid inhibition throughout a 24-h period, with the pH ≥ 4 and ≥ 5 holding time ratios for vonoprazan 20 mg bid being 100% and 99%, respectively, even in *H. pylori*-negative subjects [26]. Consistent acid inhibition increases the stability and bioavailability of acid-sensitive antibiotics, preventing their degradation and increasing their concentration in the gastric mucosa [9,27,28]. In 2016, a phase III randomized, double-blind study reported an eradication rate of 92.6% (95% CI: 89.2–95.2%) using a first-line vonoprazan-containing regimen and 75.9% (70.9–80.5%) using a lansoprazole-containing regimen [12]. Other studies have reported that the efficacy of first-line vonoprazan-containing regimens ranges from 82.9% to 94.6% [13,14,15,16,29,30,31]. Based on 21 studies that investigated a total of 12,010 patients receiving PPI-containing first-line therapy, no studies reported an eradication rate higher than 85% [32]. Therefore, as observed in this study, potent acid inhibition using vonoprazan is a key requirement for successful eradication in Japanese.

### 4.2. Eradication Rate of Vonoprazan-Based First-Line Therapy for Clarithromycin-Resistant H. pylori Strains

Clarithromycin resistance is a rising clinical problem for *H. pylori* eradication in many countries where the antibiotic has been widely used to treat patients with bacterial infection, making bacterial culture and antimicrobial sensitivity testing effective clinical options. In fact, a randomized controlled trial has shown that vonoprazan-containing first-line therapy is significantly superior to PPI-containing therapy in patients with clarithromycin-resistant strains (eradication rate: 82.0% [vonoprazan] and 40.0% [PPI], OR: 6.83, 95% CI: 3.63–12.86, *p* < 0.0001) [33]. In this study, 40.6% (56/138) of patients receiving first-line treatment were infected with clarithromycin-resistant strains and the eradication rate was 71.4% (95% CI: 57.8–82.7%) and 91.5% (95% CI: 83.1–96.5%) in patients infected with strains resistant and sensitive to clarithromycin, respectively. Infection with clarithromycin-resistant strains was identified as a predictive factor for failed eradication in multivariate analysis (OR: 5.788, 95% CI: 1.916–17.485, *p* = 0.002). Given that an eradication rate of 80% is not satisfactory, alternative treatment options, such as replacing clarithromycin with metronidazole, are needed to obtain a higher eradication rate in patients infected with clarithromycin-resistant strains. Then, future trials should investigate the efficacy of vonoprazan-containing concomitant therapy, sequential therapy, and bismuth-containing quadruple therapy. 

### 4.3. Second-Line and Third-Line Vonoprazan-Containing Eradication Therapy

While the prevalence of metronidazole-resistant strains is 5–12% in Japan and metronidazole is not an acid-sensitive antimicrobial agent, the eradication rate of PPI/metronidazole-containing second-line regimen has remained constant at approximately 80%. A meta-analysis showed that the eradication rate of vonoprazan-containing regimens (83.4%) is similar to that for PPI-containing regimens (81.2%, OR: 1.04, 95% CI: 0.77–1.42, *p* = 0.79) [34]. Therefore, PPI/amoxicillin/metronidazole regimen may be recommended as a second-line treatment over vonoprazan-containing regimens due to cost-effectiveness and comparable efficacy and safety in Japan.

Eradication therapy using PPI/amoxicillin/sitafloxacin or PPI/metronidazole/sitafloxacin is the main third-line regimen in Japan [35,36,37,38]. However, reports investigating the efficacy of vonoprazan-containing third-line treatment are limited [39]. Sitafloxacin is a new quinolone antibacterial agent that is expected to show efficacy due to its low MIC for *H. pylori*, and the low rate of sitafloxacin-resistant strains (less than 10%) is a strong motivator for its use as an eradication therapy [37,38]. Sitafloxacin is also an acid-sensitive antimicrobial agent, whose stability and bioavailability are increased by potent acid inhibition. Evidence from the present and a previous study suggest that the vonoprazan/amoxicillin/sitafloxacin regimen is more effective than PPI-containing regimens as a third-line therapy in patients [39].

### 4.4. Vonoprazan-Containing Eradication Therapy and CYP3A4/5 and CYP2C19 Genotype

Because vonoprazan is metabolized mainly by CYP3A4/5 and partially by CYP2B6, CYP2C19, and CYP2D6 [21], its pharmacokinetics and pharmacodynamics may be influenced by genetic variations in the respective genes. Many drugs are metabolized by CYP3A4, and the *CYP3A4**22 decrease-of-function allele has been associated with attenuated metabolism of *CYP3A4*-dependent drugs, such as statins and tacrolimus [40]. CYP3A4 enzyme activity in the *CYP3A4* *1 /*1 wild-type genotype is 2.5-fold higher than that in *CYP3A4* *22 carriers [40]. However, no patients in this study had the *CYP3A4* *22 allele. Therefore, the influence of this and other *CYP3A4* polymorphisms associated with CYP3A4 protein expression levels in Japanese warrants further investigation.

The inter-individual variability in the pharmacokinetics of CYP3A5-metabolized drugs is explained by a single nucleotide polymorphism in intron 3 of *CYP3A5*, 6986A > G. Blood tacrolimus levels are higher in patients with the *CYP3A5* *3/*3 genotype than *1 genotype [41,42]. We previously reported that the incidence of adverse events in ulcerative colitis patients treated with tacrolimus was significantly higher in *CYP3A5* expressers than non-expressers [43]. The elimination rate of vonoprazan was significantly correlated with CYP3A4/5 activity, suggesting that CYP3A4/5 activity influences the pharmacokinetics of vonoprazan [21]. However, although our previous preliminary report showed a significant association between *CYP3A5* genotype and vonoprazan-containing treatment outcome in the first-line triple therapy [22], when we entered the larger number of patients in this study, we failed to show a significant association. Although significant differences were not observed in the eradication rate of vonoprazan-containing regimens among CYP3A4 genotypes, the findings about the importance of clarithromycin-resistance rather than such genotype for eradication is thought to be a significance on clinical pharmacology. Previously, because there was no data to investigate pharmacokinetics (PK)/pharmacodynamics (PD) of vonoprazan among different *CYP3A5* genotypes, future trials should investigate the direct association between *CYP3A5* genotype and the PK/PD of vonoprazan to clarify this concern.

Acid inhibition by administration of vonoprazan 20 mg qd and bid is similar among *CYP2C19* genotypes [26]. Murakami et al [12] reported an eradication rate of 92.9% in *CYP2C19* EMs/IMs, which was similar to that in PMs (90.9%). We found no significant difference in eradication rates among *CYP2C19* genotypes. These observations suggest that *CYP2C19* genotype-based tailored treatment using vonoprazan may not be necessary for acid-related diseases.

## 5. Conclusions

We assessed the effect of *CYP3A4/5* and *CYP2C19* genotype and antibiotic susceptibility on clinical outcome following treatment with first-line, second-line and third-line vonoprazan-containing therapy in Japanese. Vonoprazan-related genetic variations in *CYP3A4/5* and *CYP2C19* were not associated with clinical outcome of *H pylori* eradication therapy. Although vonoprazan-containing triple therapy shows high efficacy in terms of *H. pylori* eradication compared to PPI-containing therapy, especially in patients infected with clarithromycin-resistant strains, an eradication rate of 80% is not satisfactory. We think there is potential for the development of a culture test-based tailored treatment that can achieve an eradication rate exceeding 95%.

## Figures and Tables

**Figure 1 antibiotics-09-00645-f001:**
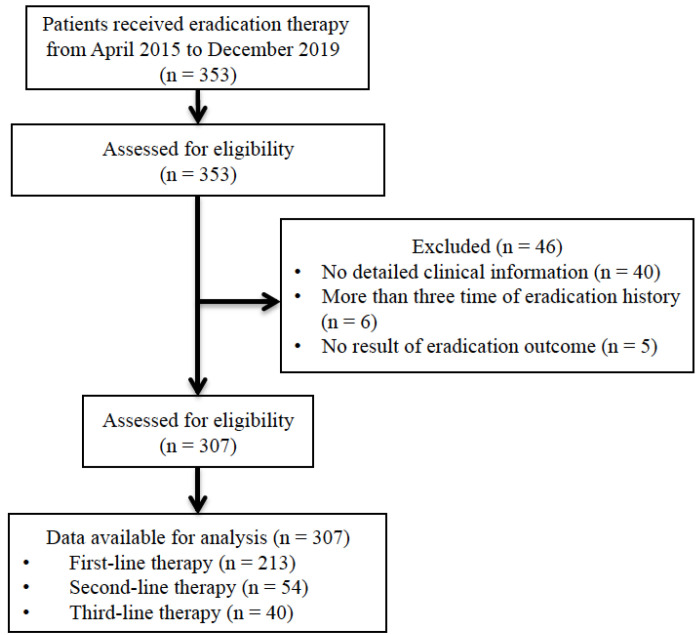
Flow for the selection of patients.

**Table 1 antibiotics-09-00645-t001:** Characteristics of patients positive for *Helicobacter pylori.*

Characteristics	Total (*n* = 307)	First-Line Regimen (*n* = 213)	Second-Line Regimen (*n* = 54)	Third-Line Regimen (*n* = 40 )	P Value
Age (years)	62.3 ± 13.1	62.8 ± 13.0	61.9 ± 13.7	69.6 ± 13.1	0.381
Sex (male/female)	160/147	109/104	32/22	19/21	0.471
Body weight (kg)	59.5 ± 11.1	59.1 ± 10.8	59.4 ± 9.1	62.1 ± 15.0	0.598
Height (cm)	161.7 ± 8.7	161.1 ± 9.0	163.5 ± 7.1	162.8 ± 8.5	0.100
Drinking (none/ past/ current)	143/17/133	105/13/87	23/3/24	15/1/22	0.471
Smoking (none/ past/ current)	188/76/29	131/53/21	30/15/5	27/8/3	0.867
*CYP2C19* genotype (UR/EM/IM/PM)	10/79/170/48	7/55/111/40	3/11/34/6	0/13/25/2	0.381
*CYP3A4* genotype (*1/*1, *1/*22, *22/*22)	307/0/0	213/0/0	54/0/0	40/0/0	1.000
*CYP3A5* genotype (*1/*1, *1/*3, *3/*3)	14/66/88	10/46/59	2/16/17	2/4/12	0.364
**Sensitivity to antimicrobial agents**					
Clarithromycin (sensitive/resistant)	92/102	82/56	7/24	3/22	< 0.001
Amoxicillin (sensitive/not sensitive/resistant)	150/44/0	118/20/0	21/10/0	11/14/0	< 0.001
Metronidazole (sensitive/resistant)	168/26	127/11	28/3	13/12	< 0.001
**Endoscopic findings**					
Gastric atrophy	1.8 ± 0.4	1.8 ± 0.4	1.8 ± 0.5	1.7 ± 0.4	0.752
Intestinal metaplasia	0.8 ± 0.7	0.9 ± 0.7	0.8 ± 0.7	0.7 ± 0.8	0.649
Enlarged folds	0.2 ± 0.4	0.2 ± 0.4	0.3 ± 0.5	0.2 ± 0.4	0.714
Nodular gastritis	0.0 ± 0.2	0.0 ± 0.2	0.1 ± 0.2	0.1 ± 0.2	0.934
Diffuse redness	1.6 ± 0.5	1.6 ± 0.5	1.6 ± 0.5	1.4 ± 0.5	0.028
Total	4.5 ± 1.3	4.5 ± 1.3	4.5 ± 1.3	4.1 ± 1.4	0.186

EM, extensive metabolizer; IM, intermediate metabolizer; PM, poor metabolizer; UM, ultra rapid metabolizer; All values are mean ± standard deviation.

**Table 2 antibiotics-09-00645-t002:** Eradication rates in the first-line, second-line and third-line regimens.

Parameters	Total (*n* = 307)	First-Line Regimen (*n* = 213)	Second-Line Regimen (*n* = 54)	Third-Line Regimen (*n* = 40)
Total eradication rate	86.3% (265/307)	84.5% (180/213)	92.6% (50/54)	87.5% (35/40)
CYP2C19 genotype				
UM (*n* = 10)	70.0% (7/10)	57.1% (4/7)	100% (3/3)	NA
EM (*n* = 79)	88.6% (70/79)	90.9% (50/55)	90.9% (10/11)	76.9% (10/13)
IM (*n* = 170)	85.3% (145/170)	81.1% (90/111)	94.1% (5/6)	92.0% (23/25)
PM (*n* = 48)	89.6% (43/48)	90.0% (36/40)	83.3% (32/34)	100% (2/2)
CYP3A5				
*1/*1 (*n* = 14)	92.9% (13/14)	90.0% (9/10)	100% (2/2)	100% (2/2)
*1/*3 (*n* = 66)	83.3% (55/66)	80.4% (37/46)	93.8% (15/16)	75.0% (3/4)
*3/*3 (*n* = 88)	85.2% (75/88)	84.7% (50/59)	94.1% (16/17)	75.0% (9/12)
Clarithromycin (sensitive/resistant)				
Sensitive	91.3% (84/92)	91.5% (75/82)	85.7% (6/7)	100% (3/3)
Resistant	78.4% (80/102) *	71.4% (40/56) *	95.8% (23/24)	77.3% (17/22)
Amoxicillin (sensitive/not sensitive/resistant)				
Sensitive	87.3% (131/150)	85.6% (101/118)	95.2% (20/21)	90.9% (10/11)
Not sensitive	75.0% (33/44)	70.0% (14/20)	90.0% (9/10)	71.4% (10/14)
Metronidazole (sensitive/resistant)				
Sensitive	85.1% (143/168)	82.7% (105/127)	92.9% (26/28)	92.3% (12/13)
Resistant	80.8% (21/26)	90.9% (10/11)	100% (3/3)	66.7% (8/12)

EM, extensive metabolizer; IM, intermediate metabolizer; PM, poor metabolizer; UM, ultra rapid metabolizer; * *p* < 0.05 vs. sensitive strains.

**Table 3 antibiotics-09-00645-t003:** Univariate and multivariate analyses of factors for eradication failure in first-line therapy.

Factor	Univariate Analysis			Multivariate Analysis		
	Odds Ratio	95% CI	*p*-Value	Odds Ratio	95% CI	*p*-Value
Male	1.830	0.849–3.940	0.123	3.329	1.149–9.645	0.027
Age	0.994	0.967–1.022	0.662			
Smoking, past	1.048	0.428–2.566	0.918			
Smoking, current	0.982	0.264–3.661	0.979			
Drinking, current	1.151	0.552–2.538	0.728			
Endoscopic gastric atrophy	1.593	0.558–4.552	0.384			
Endoscopic intestinal metaplasia	1.445	0.861–2.426	0.163	1.193	0.589–2.417	0.625
Endoscopic diffuse redness	2.868	1.092–7.530	0.032	0.882	0.278–2.803	0.832
Endoscopic total score	1.470	1.057–2.043	0.022			
CYP2C19 IM (vs. CYP2C19 EM	1.575	0.652–3.803	0.313			
CYP2C19 PM (vs. CYP2C19 EM)	0.750	0.210–2.677	0.710			
CYP3A5 *3/*3 type (vs. *1 carrier)	0.828	0.309–2.219	0.707			
Clarithromycin -R (vs. clarithromycin-S)	4.286	1.628–11.279	0.003	5.788	1.916–17.485	0.002
Amoxicillin-NS (vs. amoxicillin-S)	2.546	0.860–7.541	0.092	2.593	0.759–8.851	0.128

CI, confidence interval; EM, extensive metabolizer of CYP2C19; IM, intermediate metabolizer of CYP2C19; PM, poor metabolizer of CYP2C19; R, resistant; S, sensitive; NS, not sensitive.

**Table 4 antibiotics-09-00645-t004:** Adverse events related to eradication treatment.

Parameters	Total(*n* = 260)	First-LineRegimen(*n* = 181)	Second-LineRegimen(*n* = 47)	Third-LineRegimen(*n* = 32)	P value
None	197 (75.8%)	141 (77.9%)	33 (70.2%)	23 (71.9%)	0.472
Diarrhea	19 (7.3%)	11 (6.1%)	4 (8.5%)	4 (12.5%)	
Loose stool	18 (6.9%)	8 (4.4%)	8 (17.0%)	2 (6.2%)	
Abdominal pain	9 (3.5%)	6 (3.3%)	1 (2.1%)	2 (6.2%)	
Allergic reaction	10 (3.8%)	9 (5.0%)	1 (2.1%)	0 (0%)	
Other	7 (2.7%)	6 (3.3%)	0 (0%)	1 (3.1%)

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
