# Peer review of "Effect of Antibiotic Susceptibility and CYP3A4/5 and CYP2C19 Genotype on the Outcome of Vonoprazan-Containing Helicobacter pylori Eradication Therapy"

_antibiotics, 2020, doi:10.3390/antibiotics9100645_

Round 1

Reviewer 1 Report

Good work ! The manusript is also weel written. Just minor revision:

2.Material and Methods - You could add a Study flow chart as figure -

Line 103: you used a commercial kit for the real time PCR assay, or what ?

please, add the name of this assay ... "TaqMan, Applied Biosystem" is already the full name or not ?

References - Check the style of all references

Author Response

Our responses to comments raised by Reviewer 1

  1. Material and Methods - You could add a Study flow chart as figure

Response: Thank you for your suggestion. We added a study flow chart as figure 1 in this revised version.

  1. Line 103: you used a commercial kit for the real time PCR assay, or what?

Response: In this study, the genotyping was performed with the use of an Applied Biosystems StepOnePlus real-time PCR system. We added this information in the revised version.

  1. Please, add the name of this assay ... "TaqMan, Applied Biosystem" is already the full name or not?

Response:

As same as response of above question 2, we deleted "TaqMan, Applied Biosystem" and revised this as "StepOnePlusTM, Applied Biosystems" in the revised version.

  1. References - Check the style of all references

Response: In revised version, we checked the style of all references.

Reviewer 2 Report

The manuscript entitled “Effect of antibiotic susceptibility and CYP3A4/5 and

CYP2C19 genotype on the outcome of vonoprazan-containing Helicobacter pylori eradication therapy” describes that H. pylori-infected patients with genotyping for CYP3A4 *1/*22, CYP3A5 *1/*3 and CYP2C19 *1/*2/*3/*17 before vonoprazan-containing eradication treatment may not be useful for predicting clinical outcomes. Authors assess the effect of CYP3A4/5 and CYP2C19 genotype and antibiotic susceptibility on clinical outcome following treatment with first-line, second-line and third-line vonoprazan-containing therapy in Japanese. Vonoprazan-related genetic variations in CYP3A4/5 and CYP2C19 are not associated with clinical outcome of H pylori eradication therapy. Although vonoprazan-containing triple therapy shows high efficacy in terms of H. pylori eradication compared to PPI-containing therapy, especially in patients infected with clarithromycin-resistant strains, an eradication rate of 80% is not satisfactory. We think there is potential for the development of a culture test-based tailored treatment that can achieve an eradication rate exceeding 95%.

Antibiotics resistance of H. pylori is still a major problem for H. pylori eradication and subsequent gastric diseases such as gastritis, gastric ulcer, duodenal ulcer precancerous change and gastric cancer. In the Type-II DM study, successful H. pylori eradication can improve insulin sensitivity. So, this study is useful for H. pylori eradication therapy, gastric cancer prevention and Type-II DM improvement.

Author Response

Our responses to comments raised by Reviewer 2

  1. Antibiotics resistance of H. pylori is still a major problem for H. pylori eradication and subsequent gastric diseases such as gastritis, gastric ulcer, duodenal ulcer precancerous change and gastric cancer. In the Type-II DM study, successful H. pylori eradication can improve insulin sensitivity. So, this study is useful for H. pylori eradication therapy, gastric cancer prevention and Type-II DM improvement.

Response:

Thank you for your comments. We also believe this study is useful for H. pylori eradication therapy, gastric cancer prevention and Type-II DM improvement, as your comments.

Reviewer 3 Report

The aim of the work entitled “Effect of antibiotic susceptibility and CYP3A4/5 and CYP2C19 genotype on the outcome of vonoprazan-containing Helicobacter pylori eradication therapy” was to determine the effect of vonoprazan-associated genetic variations in CYP3A4/5 and CYP2C19 on the outcome of antibiotic eradication against H. pylori.

I believe that the manuscript is carefully written and worth publishing in a journal. I only have some minor suggestions:

- “A total of 307 Japanese who were genotyped for CYP3A4 *1/*22, CYP3A5 *1/*3 and CYP2C19 *1/*2/*3/*17 and investigated for susceptibility to antimicrobial agents received vonoprazan-containing regimens …” -> A total of 307 Japanese who were genotyped for CYP3A4 *1/*22, CYP3A5 *1/*3 and CYP2C19 *1/*2/*3/*17, and investigated for susceptibility to antimicrobial agents, received vonoprazan-containing regimens …” [Abstract]

- Please transfer part of Table 1 to the other page (all should be on one page)

- Table 1 and 2: please explain the differentiation of amoxicillin sensitivity (are there two or three categories?): if there are three categories (sensitive/non sensitive/resistant), how do they differ in the MIC values?

- Please standardize the references, i.e. add DOI to each reference

Author Response

Our responses to comments raised by the Reviewer 3

  1. - “A total of 307 Japanese who were genotyped for CYP3A4 *1/*22, CYP3A5 *1/*3 and CYP2C19 *1/*2/*3/*17 and investigated for susceptibility to antimicrobial agents received vonoprazan-containing regimens …” -> A total of 307 Japanese who were genotyped for CYP3A4 *1/*22, CYP3A5 *1/*3 and CYP2C19 *1/*2/*3/*17, and investigated for susceptibility to antimicrobial agents, received vonoprazan-containing regimens …” [Abstract]

Response:

We revised “A total of 307 Japanese who were genotyped for CYP3A4 *1/*22, CYP3A5 *1/*3 and CYP2C19 *1/*2/*3/*17 and investigated for susceptibility to antimicrobial agents received vonoprazan-containing regimens …” in ”A total of 307 Japanese who were genotyped for CYP3A4 *1/*22, CYP3A5 *1/*3 and CYP2C19 *1/*2/*3/*17, and investigated for susceptibility to antimicrobial agents, received vonoprazan-containing regimens …” in the revised version.

  1. - Please transfer part of Table 1 to the other page (all should be on one page)

Response:

Thank you for your comments. In this revised version, we transfer part of Table 1 to the other page for setting on one page.

  1. - Table 1 and 2: please explain the differentiation of amoxicillin sensitivity (are there two or three categories?): if there are three categories (sensitive/non sensitive/resistant), how do they differ in the MIC values?

Response:

The cut-off MICs used to define resistance and the absence of sensitivity were > 0.5 µg/mL and > 0.06 µg/mL for amoxicillin, respectively. According to your suggestion, we added differences of three categories for the MIC values.

  1. - Please standardize the references, i.e. add DOI to each reference

Response:

In revised version, we checked the style of all references. However, any references have no DOI.